# Biophysical basis for brain folding and misfolding patterns in ferrets and humans

Gary PT Choi[1], Chunzi Liu[2], Sifan Yin[2], Gabrielle Séjourné[2,3], Richard S Smith[4], Christopher A Walsh[5,6], L Mahadevan[2,7,8]*

[1]Department of Mathematics, The Chinese University of Hong Kong, Hong Kong, China; [2]School of Engineering and Applied Sciences, Harvard University, Cambridge, United States; [3]Department of Cell Biology, Duke University, Durham, United States; [4]Department of Pharmacology, Feinberg School of Medicine, Northwestern University, Chicago, United States; [5]Division of Genetics and Genomics, Manton Center for Orphan Disease, and Howard Hughes Medical Institute, Chevy Chase, United States; [6]Boston Children's Hospital, Boston, United States; [7]Department of Organismic and Evolutionary Biology, Harvard University, Cambridge, United States; [8]Department of Physics, Harvard University, Cambridge, United States

*For correspondence:
lmahadev@g.harvard.edu

Competing interest: The authors declare that no competing interests exist.

## eLife Assessment

This **important** study characterizes the morphogenesis of cortical folding in the ferret and human cerebral cortex using complementary physical and computational modeling. Notably, these approaches are applied to charting, in the ferret model, known abnormalities of cortical folding in humans. The study finds **convincing** evidence that variation in cortical thickness and expansion accounts for deviations in morphology and supports these findings using cutting-edge approaches from both physical gel models and numerical simulations. The study will be of broad interest to the field of developmental neuroscience.

**Abstract** A mechanistic understanding of neurodevelopment requires us to follow the multiscale processes that connect molecular genetic processes to macroscopic cerebral cortical formations and thence to neurological function. Using MRI of the brain of the ferret, a model organism for studying cortical morphogenesis, we create in vitro physical gel models and in silico numerical simulations of normal brain gyrification. Using observations of genetically manipulated animal models, we identify cerebral cortical thickness and cortical expansion rate as the primary drivers of dysmorphogenesis and demonstrate that in silico models allow us to examine the causes of aberrations in morphology and developmental processes at various stages of cortical ontogenesis. Finally, we explain analogous cortical malformations in human brains, with comparisons with human phenotypes induced by the same genetic defects, providing a unified perspective on brain morphogenesis that is driven proximally by genetic causes and affected mechanically via variations in the geometry of the brain and differential growth of the cortex.

## Introduction

Understanding the growth and form of normal and abnormal cortical convolutions (gyri and sulci) is important for the study of human neurodevelopmental diseases (*Lui et al., 2011*; *Geschwind and Rakic, 2013*; *Molnár et al., 2019*; *Del-Valle-Anton and Borrell, 2022*; *Akula et al., 2023a*). During early brain development, the cortical plate expands tangentially relative to the underlying white

**eLife digest** The wrinkled and folded surface of the human brain is both iconic and familiar. These folds allow a large cortical surface area to fit inside the skull and are essential for healthy brain function. The folds form during development when the brain's outer layer – the cerebral cortex – grows faster than the tissue beneath it, causing the surface to buckle.

When this process is disrupted, the brain can develop abnormal folding patterns known as malformations of cortical development. In humans, these conditions are associated with epilepsy, intellectual disability and developmental delay. Studying how such malformations arise is challenging because human brain folding occurs before birth.

To address this, researchers use animal models. The ferret is particularly valuable because its brain develops folds similar to those in humans, and many genes linked to human cortical malformations produce comparable folding defects in ferrets. Choi et al. wanted to find out whether the wide range of brain folding abnormalities seen in ferrets and their homologs in humans could be explained by changes in just a few physical properties of the developing brain. Specifically, they tested whether mutations linked to human cortical malformations alter the thickness or growth rate of the cortex. This question is important because different genetic syndromes often result in surprisingly similar brain shapes.

By combining brain imaging, computer simulations, and physical gel models of brains that fold when their surfaces absorb solvents and swell (similar to how fingertips swell and wrinkle when wetted for a while), Choi et al. showed that normal brain folding in ferrets can be explained by mechanical forces generated during cortical growth.

Both physical experiments with gels and computer simulations of brains with varying cortical thickness or growth rates reproduced folding patterns seen in both healthy and genetically altered ferret brains and their human homologs. Local thinning of the cortex generated many small, tightly packed folds, resembling polymicrogyria, a condition linked to mutations such as those affecting the gene *SCN3A*, which encodes instructions to form a sodium channel. Reducing overall growth produced smaller, less folded brains similar to microcephaly, which is associated with genes such as *ASPM*. In contrast, weaker folding with shallow grooves – characteristic of lissencephaly – emerged when growth was reduced and cortical thickness increased, as seen with disruptions to genes such as *TMEM161B*.

These results suggest that diverse human genetic disorders converge on common physical mechanisms that shape the brain. The work of c provides a unifying framework linking specific genes to brain shape through physical growth processes. The ferret provides a useful model organism with direct implications for human brain development and misfolding. In the future, it could help researchers interpret human brain scans and understand why different genetic disorders lead to similar malformations. However, before such insights can inform clinical practice, the models will need to incorporate additional biological detail and examine how altered folding affects brain function. More broadly, the paper also raises the question of how variations in brain folding patterns arise in non-human brains, which is the subject of a related study.

matter (**Welker, 1990**). This pattern of growth is the central cause of gyrification; indeed, tangential cortical expansion creates compressive forces on the faster-growing outer layer of the cortex and tensile forces on the attached slower-growing inner layer, and the relative-growth induced forces cause cortical folding as suggested more than a century ago (**His, 1868**) and first quantitatively elucidated nearly 50 years ago (**Richman et al., 1975**). At a molecular and cellular level, neurogenesis, neuronal migration, and neuronal differentiation all contribute to the tangential growth of the developing cortex via processes such as an increase in either the number or size of cells (**Fietz et al., 2010**; **Hansen et al., 2010**; **Borrell, 2018**; **Van Essen, 2023**). Recent models that take these facts into account attempt to explain gyrification in terms of a simple mechanical instability, termed sulcification (**Hohlfeld and Mahadevan, 2011**), that, when iterated with variations (**Tallinen et al., 2013**), shows that tangential expansion of the gray matter constrained by the white matter can explain a range of different morphologies seen in the brains of different organisms (**Tallinen et al., 2014**; **Kroenke and Bayly, 2018**). Furthermore, when deployed over developmental time to simulate normal human

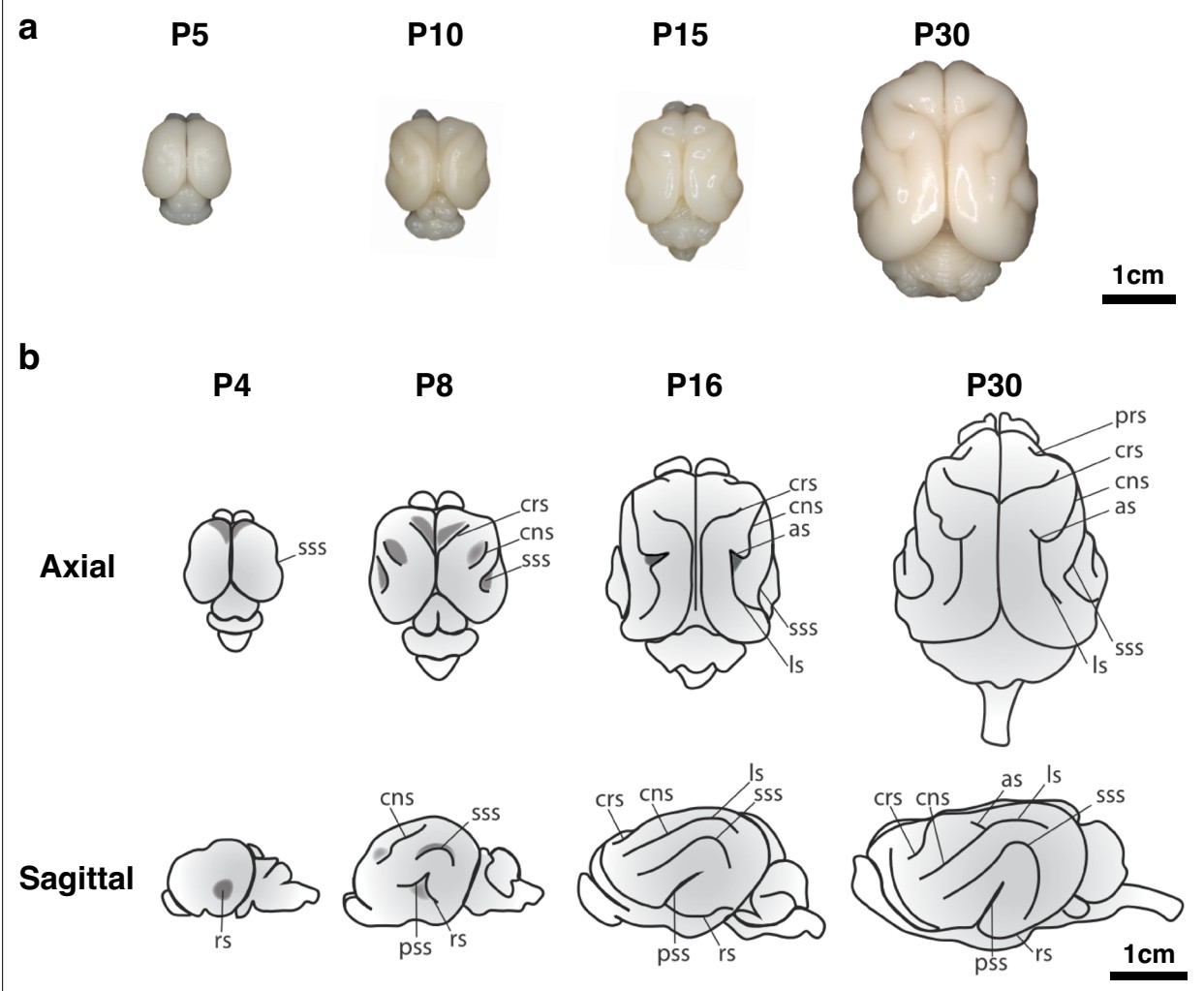

**Figure 1.** Time course of ferret brain morphogenesis. (**a**) Whole brain samples from ferrets of various ages show progressive development of cortical gyri and sulci. (**b**) Ferret brains show an increase in complexity of sulcal pattern and in sulcal depth throughout development. The rhinal sulcus (rs), cruciate sulcus (crs), coronal sulcus (cns), suprasylvian sulcus (sss), pseudosylvian sulcus (pss), lateral sulcus (ls), and ansate sulcus (as) are labeled. Schematic by G. Séjourné.

cortical convolution, the results can capture a substantial range of features seen in normal human fetal brain morphogenesis (*Tallinen et al., 2016*). However, these and other similar studies (*Toro and Burnod, 2005*; *Toro et al., 2008*; *Herculano-Houzel, 2009*; *Azevedo et al., 2009*; *Hutton et al., 2009*; *Nie et al., 2010*; *Bayly et al., 2013*; *Budday et al., 2014*; *Garcia et al., 2021*; *Heuer et al., 2023*; *Pang et al., 2023*; *Schwartz et al., 2023*) do not allow us to understand malformations of cortical development (MCD), neurodevelopmental disorders that result from disrupted human cerebral cortex formation during embryonic brain development (*Desikan and Barkovich, 2016*), owing to our inability to probe the development of the human fetal brain in utero. In addition, MCDs are difficult to relate to the physical properties of cerebral cortical folding because of the uncertainties in defining the effects of specific human genetic abnormalities on specific cortical features such as thickness and surface area.

An alternative strategy is to turn to model organisms to study the developmental trajectory of MCDs. Since commonly used animal models such as the mouse and rat have lissencephalic cortices, the ferret, a gyrencephalic non-primate, has been favored as an experimentally tractable laboratory organism that demonstrates cortical folding patterns that are roughly similar to that observed in the human (*Neal et al., 2007*; *Fietz et al., 2010*; *Sawada and Watanabe, 2012*; *Johnson et al., 2018*; *Gilardi and Kalebic, 2021*). Furthermore, since the process of cortical folding in the ferret is

almost exclusively postnatal, with the progressive development of cortical gyri and sulci from postnatal day 0 (P0) to adolescence (*Figure 1*), it is more easily observable. Finally, the ability to perform region-specific genetic manipulation of the ferret brain through in utero electroporation (*Masuda et al., 2015*; *Tabata and Nakajima, 2001*) makes the ferret an ideal system for modeling normal and abnormal neurodevelopmental processes.

Inspired by our previous studies using physical experiments with swelling gels and computational models of brain growth (*Tallinen et al., 2014*; *Tallinen and Biggins, 2015*; *Tallinen et al., 2016*), we model the folding of a normal ferret brain using a physical gel model and a computational model based on the principle of constrained cortical expansion and compare the simulation results with the real brain development using various geometric morphometric approaches. We then use the computational and physical models to reproduce defective developmental processes of the ferret brain and show that they are consistent with biological experiments that manipulate different molecular drivers of neurogenesis, neuronal migration, and cell growth in the cortex that underlie its relative thickness and expansion rate. Taken together, our studies provide a mesoscopic approach to brain morphogenesis that combines computational in silico and physical gel in vitro models with morphological and molecular analysis of ferret cortical disease models and shed light on analogous MCDs in human brains.

## Results

### Physical gel model

Inspired by the observation that soft physical gels swell and fold superficially when immersed in solvents, we constructed a physical simulacrum of ferret brain folding following our previous protocols (*Tallinen et al., 2014*; *Tallinen et al., 2016*). Specifically, we produced two-layer PDMS gel models of the ferret brain at various ages based on surfaces reconstructed from MR images (see Appendix 1 for details). We then immersed the two-layer gel brain model in n-hexane, which led to folding patterns by solvent-driven swelling of the outer layers (*Figure 2a*).

*Figure 2b* shows the experimental results for a P8 gel brain; it swells nonuniformly and folds progressively from an initial state that has invaginations corresponding to the cruciate sulcus (crs), the coronal sulcus (cns), and the suprasylvian sulcus (sss). The post-swelling state shows the development of sulci corresponding in location and self-contacting nature to the crs, cns, sss, and the formation of rhinal sulcus (rs), the pseudosylvian sulcus (pss), and lateral sulcus (ls), and ansate sulcus (as) observed in real ferrets aged P21 and older. *Figure 2c* shows another swelling experiment with a starting shape corresponding to the P16 gel brain, from which we observe a similar progression in the folding patterns. We see that our minimal physical model can capture the qualitative aspects of the folding transitions in the ferret brain (see also Appendix 1 and *Videos 1 and 2*).

### Computational model

To complement our physical experiments with quantitative simulations of ferret brain development, we followed the approach in *Tallinen et al., 2014*; *Tallinen et al., 2016* and considered a neo-Hookean material model for the brain cortex consisting of a layer of gray matter on top of a deep layer of white matter with volumetric strain energy density

$$W = \frac{\mu}{2}\left[Tr(\mathbf{F}\mathbf{F}^T)J^{-2/3} - 3\right] + \frac{K}{2}(J-1)^2, \tag{1}$$

where $\mathbf{F}$ is the deformation gradient, $J = \det(\mathbf{F})$, $\mu$ is the shear modulus, and $K$ is the bulk modulus. We assume that $K = 5\mu$ for a modestly compressible material. Computer simulations were then performed on tetrahedral meshes of ferret brains to model the gyrification (see Appendix 1 for more details).

We considered both simulations that modeled the changes in brain morphology from P0 to P32 as one continuous process (*Figure 3a*, see also *Video 3*) and stepwise simulations that considered the growth process in stages, that is, from P0 to P4, from P4 to P8, from P8 to P16 and from P16 to P32 (*Figure 3b*, see also *Video 4*). In both sets of simulations, the emergence of cortical folding can be observed. In the continuous simulation approach, we observed the appearance of multiple minor folds since the continuous simulations only depend on the P0 initial brain, so that the effect of minor features in the P0 brain on the brain growth may accumulate over time. By contrast, in the stepwise

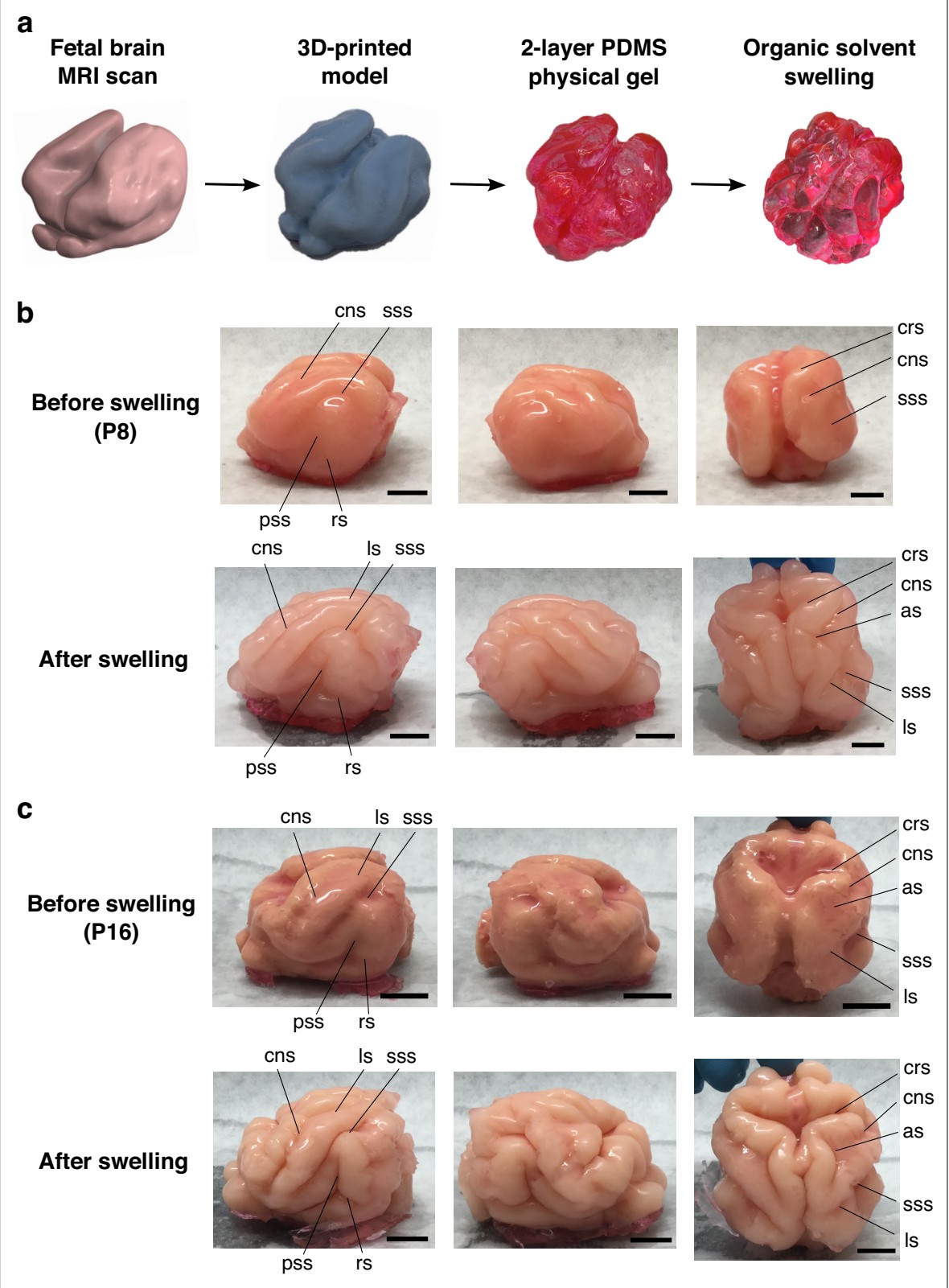

**Figure 2.** Physical gel model of ferret brain morphogenesis. (**a**) Schematic of the gel experiment. We first produced a two-layer gel model of a ferret brain from MRI scans as previously described (*Tallinen et al., 2016*). We then immersed the gel model in n-hexane for 1.5 hours, which induced the outer layer to swell by absorbing the solvent over time, resulting in the development of cortical gyri and sulci. (**b**) The swelling experiment for the P8 ferret model, in which it can be observed that the swelling of the cortical layer produces sulcal patterns and characteristics comparable to the real ferret

*Figure 2 continued on next page*

*Figure 2 continued*

brain. (**c**) The swelling experiment for the P16 gel model. Scale bar = 1 cm. Notation guide: cruciate sulcus (crs), coronal sulcus (cns), suprasylvian sulcus (sss), rhinal sulcus (rs), pseudosylvian sulcus (pss), lateral sulcus (ls), and ansate sulcus (as).

simulation approach, which focuses on multiple shorter growth periods and thus reduces the accumulation of shape variations over time. Comparing the P16 results of stepwise numerical simulation, the gel experiment, and the P16 real brain, we observe that the folding patterns are visually very similar (*Figure 4a*). For a more quantitative comparison, we applied a method based on aligning landmarks using spherical mapping termed FLASH (Fast Landmark Aligned Spherical Harmonic Parameterization) (*Choi et al., 2015*) to parameterize the simulated P16 ferret brain and the P16 brain surface generated from the MRI scans onto the unit sphere using landmark-aligned optimized conformal mappings, with the coronal sulcus (cns), suprasylvian sulcus (sss), presylvian sulcus (prs), and pseudosylvian sulcus (pss) on both the left and right hemispheres used as landmarks (see *Figure 4a* and Appendix 1 for more details). We then assessed the geometric similarity of the two brain surfaces on the spherical domain in terms of their shape index (*Koenderink and van Doorn, 1992*), which is a surface measure defined based on the surface mean curvature and Gaussian curvature. The similarity of the two shape indices suggests that the folding pattern produced by our simulation is close to the actual folding pattern (see Appendix 1 for additional analyses and *Yin et al., 2025* for more details of the morphometric method). We further utilized spherical harmonic-based representations for comparing the real and simulated ferret brains at different maximum orders, which also show that they have consistent geometric similarities (see Appendix 1).

## Physical gel and computational models for misfolding ferret brains

The effectiveness of our differential growth-based model in quantifying the normative development of normal ferret brains begs the question of whether we can use the same framework to study MCD. Here, we first consider simulating ferret cortical misfolding using both modified physical gel and computational models. In *Figure 5a*, we performed a modified numerical experiment on the P8 brain with the cortical thickness reduced to 1/4 of the original thickness globally. We also performed a modified gel brain experiment by surface-coating 1 layer of PDMS gel onto the core layer instead of 4 layers as in the original gel model in *Figure 2*, equivalent to reducing the cortical layer thickness to 1/4 of the original one. In both the numerical and gel experimental results, it can be observed that small, tightly packed folds are formed. In *Figure 5b*, we performed another modified numerical experiment with the cortical thickness doubled globally. We also performed a modified gel brain experiment by surface-coating eight layers of PDMS gel onto the core layer instead of four layers as in the original gel model to double the cortical layer thickness. In both the numerical and gel experimental results, it can be observed that the number of small folds is significantly reduced. In both cases, the numerical and gel results show a good qualitative match.

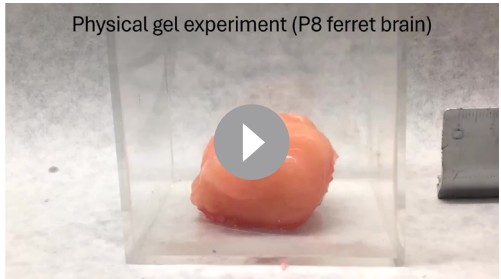

**Video 1.** Time-lapse video of the swelling experiment for the P8 gel brain, where the 2-layer PDMS model was immersed in n-hexane for 1.5 hours.

https://elifesciences.org/articles/107141/figures#video1

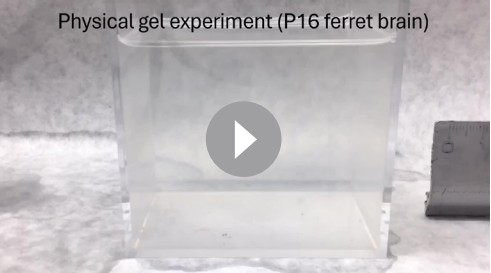

**Video 2.** Time-lapse video of the swelling experiment for a P16 gel brain, where the two-layer PDMS model was immersed in n-hexane for 1.5 hours.

https://elifesciences.org/articles/107141/figures#video2

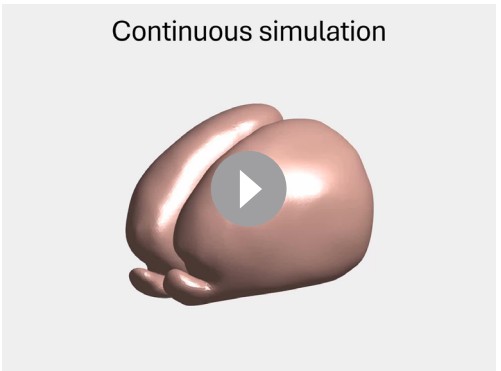

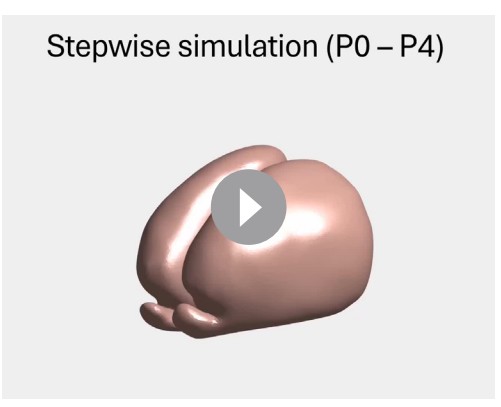

**Video 3.** Continuous numerical simulation of the ferret brain folding from P0 to P32.
https://elifesciences.org/articles/107141/figures#video3

**Video 4.** Stepwise numerical simulations of the ferret brain folding from P0 to P4, from P4 to P8, from P8 to P16, and from P16 to P32.
https://elifesciences.org/articles/107141/figures#video4

## Neurology of ferret and human cortical malformations

After performing the modified physical gel and computational experiments for ferret brain misfoldings, we proceed to consider the effect of human genetic variants on brain gyrification in ferrets in prior studies (see *Table 1*). For instance, *SCN3A* encodes a sodium channel, and specific missense mutations of it are associated with the MCD polymicrogyria (PMG) (*Smith et al., 2018*), while genetic manipulations of *ASPM* have also been shown to produce severe microcephaly in human and ferret brains (*Johnson et al., 2018*), while disrupting *TMEM161B* leads to cortical malformations in humans and ferrets (*Akula et al., 2023b*). In these and other examples, various genetic causes lead to variations in the cortical thickness ratio $h/R$ and/or the tangential growth ratio $g$ in space and time, which we know to be critical geometric parameters that change the physical nature of the sulcification instability driving cortical folding.

In *Figure 6a*, we show a control human brain MRI (top), a wild-type P16 ferret brain MRI (middle), and the stepwise numerical simulation result under the normal parameter setup (bottom). In *Figure 6b*, (top) we show the MCD PMG phenotype in humans associated with overexpression of a mutated *SCN3A* gene (*Smith et al., 2018*). In *Figure 6b* (middle), we show that the same mutation in ferrets leads to an increased number of tightly packed folds in the perisylvian region (*Smith et al., 2018*). To model this cortical malformation, we performed a modified numerical simulation with the cortical thickness reduced to 1/4 of the original thickness at a localized zone around the perisylvian region of the P8 brain model. From the modified numerical simulation result (*Figure 6b*, bottom), we can see that our model can qualitatively capture perisylvian PMG (see also Appendix 1).

In *Figure 6c* (top) and *Figure 6c* (middle) (*Johnson et al., 2018*), we show that *ASPM* mutants produce severe microcephaly in human and ferret brains, because the cortical surface area is reduced while there is no significant change in the cortical thickness. To reproduce this malformation, we consider a modified computational experiment with a reduction of the growth rate to 1/4 of the original rate. In *Figure 6c* (bottom), we show the results of numerical simulations that lead to less prominent folding when compared to the normal brain, consistent with observations in human and ferret brains.

Finally, in *Figure 6d* (top) and *Figure 6d* (middle) (*Akula et al., 2023b*), we show that disrupting *TMEM161B* also leads to cortical malformations in human and ferret brains, with shallower sulci. Modifying the morphogenetic simulations with a reduction of the growth rate to 3/4 of the original rate and an increase of the cortical thickness to 1.5 times the original thickness leads to results shown in *Figure 6d* (bottom) that match the experimentally observed malformation patterns qualitatively.

In Appendix 1, we present additional computational experiments to demonstrate the effect of different combinations of the growth rate and cortical thickness parameters on the cortical malformation results.

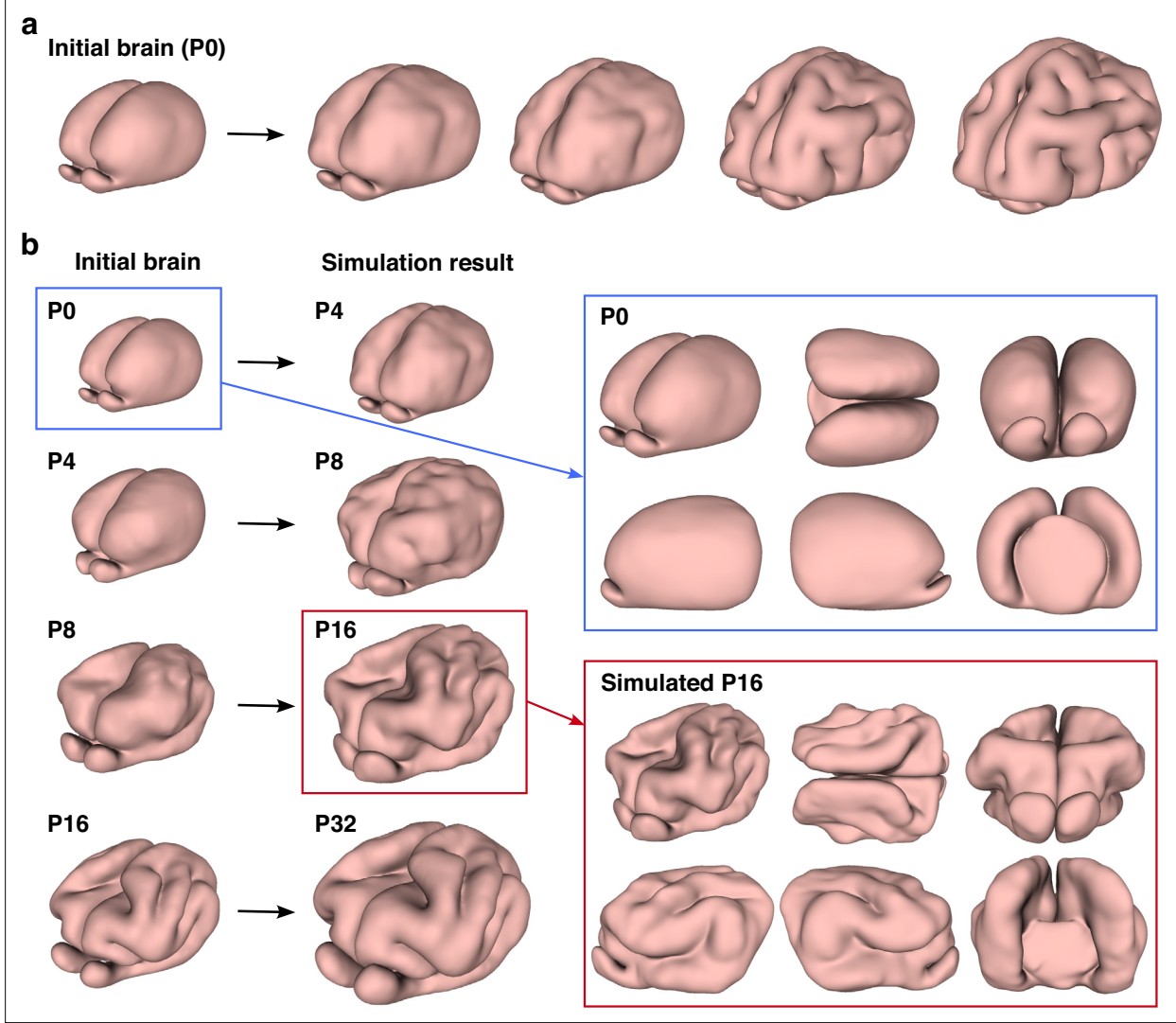

**Figure 3.** Numerical model of ferret brain morphogenesis. (**a**) Continuous growth simulation from P0 to adolescence. The P0 brain tetrahedral mesh was used as the input for the numerical simulation. (**b**) Stepwise growth simulation from P0 to P4, P4 to P8, P8 to P16, and P16 to P32. For different growth intervals, we use different brain tetrahedral meshes as the input for the numerical simulation. Different views of the input P0 brain and the simulated P16 brain are provided.

## Discussion

Understanding the growth and form of the cerebral cortex is a fundamental question in neurobiology, and the experimentally accessible progressive postnatal development of the ferret brain makes it an ideal system for analysis. Here, we have used a combination of physical and computational models based on differential growth to show how ferret brain morphologies arise. One may also use the physical and computational models to compare the ferret brain folding patterns with those in macaques and humans, as in our companion study (*Yin et al., 2025*).

By modifying the scaled cortical layer thickness and the tangential growth profile in our model, we have qualitatively reproduced various cortical malformations and shown how developmental mechanisms lead to morphological manifestations with potential functional implications. As several diseases, such as ion channels (*Smith and Walsh, 2020*; *Smith et al., 2021*), converge on brain malformations, future studies to validate across disease pathways could leverage these results. Altogether, our study elucidates the normal and abnormal folding in the ferret brain as a function of its genetic antecedents that lead to changes in the geometry of the cortex and thence to different physical folding patterns with functional consequences. A computational and physical-gel brain study informed by detailed

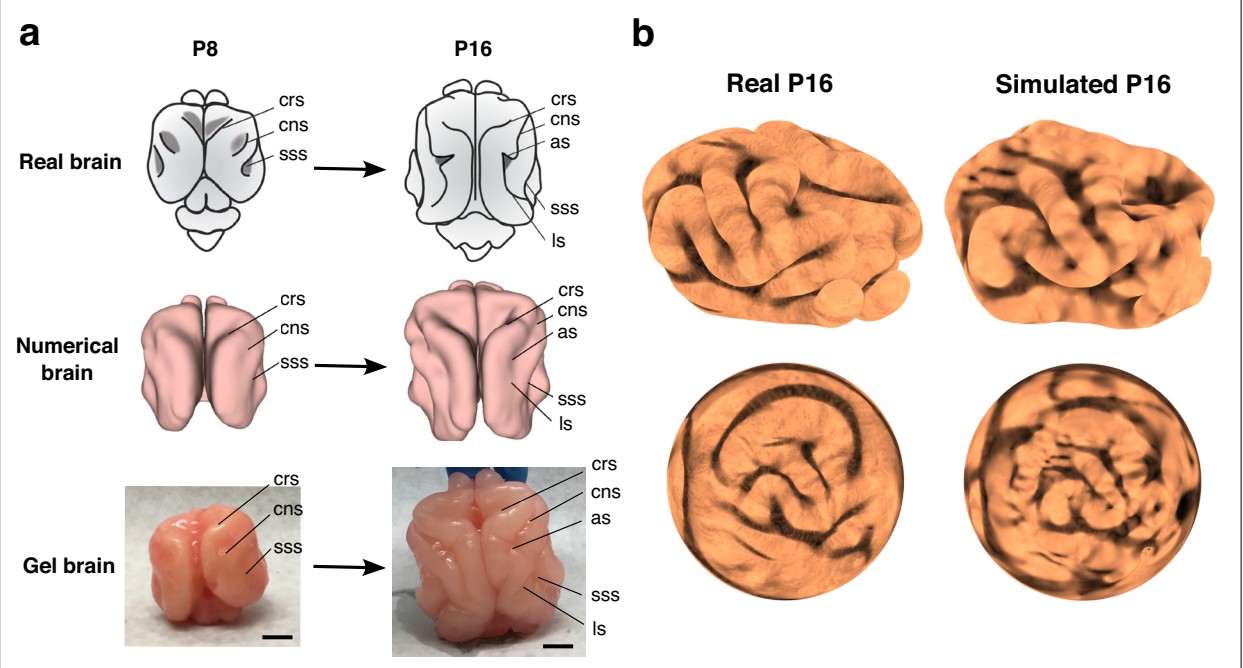

**Figure 4.** Comparison of cortical folding in real and simulated ferret brain models. (**a**) The top row shows the increase in complexity of sulcal pattern and in sulcal depth of ferret brains from P8 to P16. The middle row shows a numerical model of a P8 brain and its deformed state mimicking progression to P16. The bottom row shows a physical gel model of P8 and its post-swelling state mimicking progression to P16 (scale bar = 1 cm). The P8 initial states have invaginations corresponding to the cruciate sulcus (crs), coronal sulcus (cns), and suprasylvian sulcus (sss), and both the numerical deformed state and the physical post-swelling state have sulci corresponding in location and self-contacting nature to the crs, cns, sss, lateral sulcus (ls), and ansate sulcus (as) observed in P16 real ferrets. (**b**) The real P16 brain reconstructed from MRI scans, the simulated P16 brain, and their respective landmark-aligned spherical mappings obtained by the FLASH algorithm (*Choi et al., 2015*), each color-coded with the shape index (*Koenderink and van Doorn, 1992*) of the brain.

MRI of ferret and human fetal brains allows us to move towards a synthesis of the genetic, physical, and morphological basis for cortical malformations. Natural next steps include accounting for varying spatio-temporal expansion rates of the cortex to capture the quantitative differences in the development of fetal folding patterns in both ferrets and humans, and understanding the functional consequences as a result of impaired connectivity due to misfolding.

## Materials and methods
### Physical gel model for ferret brain folding
Beginning with T2-weighted motion-corrected anatomical MR images of ferret brains of various ages (*Toro et al., 2018b*), digital maps of the surfaces of native (pre-swollen) brain were recreated. Then, we followed our prior experimental approach (*Tallinen et al., 2014*; *Tallinen et al., 2016*) and produced two-layer PDMS gel models of the ferret brain at various ages based on the reconstructed brain surfaces. Specifically, we first generated a negative rubber mold with Ecoflex 00-30 from a 3D-printed brain plastic model and then the core gel with SYLGARD 184 at a 1:45 crosslinker:base ratio. To mimic the cortical layer, we surface-coated four layers of PDMS gel at a 1:35 crosslinker:base ratio onto the core layer. Finally, tangential cortical growth was mimicked by immersing the two-layer gel brain model in n-hexane for 1.5 hours, which resulted in solvent-driven swelling of the outer layers, leading to folding patterns. See Appendix 1 for details.

### Computational model for ferret brain folding
Three geometrical parameters of the 3D brain models control its morphogenesis: the average brain size $R$ (determined for example by its volume), the average cortical thickness $T$, and the average tangential expansion ratio of the cortex relative to the white matter, $g^2$. To characterize brain development

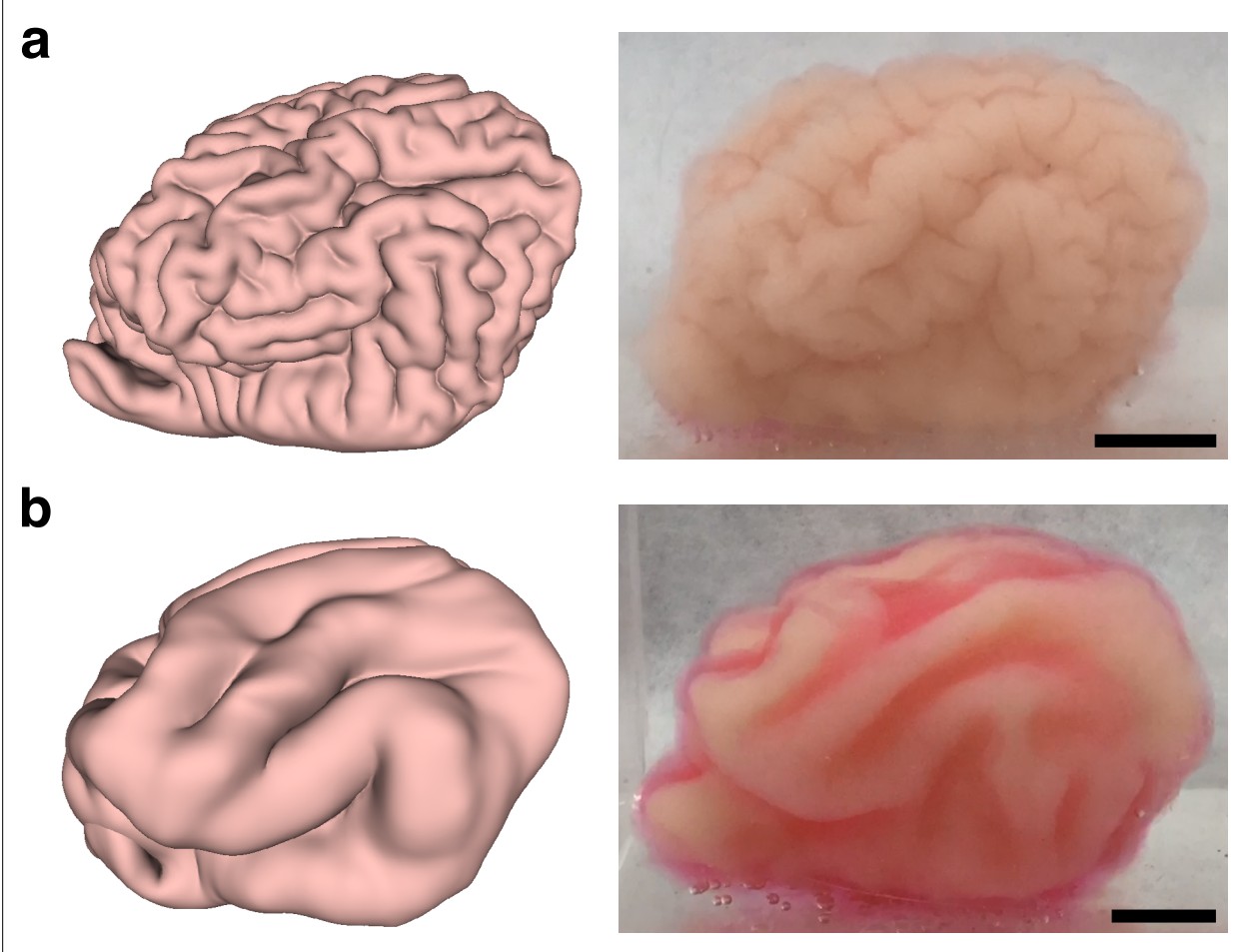

**Figure 5.** Numerical and gel experiments on the P8 ferret brain with globally modified brain gyrification. (**a**) The numerical and gel experimental results with a global reduction of the cortical thickness to 1/4 of the original thickness. (**b**) The numerical and gel experimental results with a global increase in the cortical thickness to twice the original thickness. Scale bar = 1 cm.

in the ferret, we followed the empirical scaling laws for gray matter volume to thickness described in *Tallinen et al., 2014* and set $R/T \approx 10$ with the tangential expansion ratio $g \approx 1.9$, along with an indicator function $\theta(y) = (1 + e^{10(y/T-1)})^{-1}$, with $y$ the distance from surface in a material reference frame used to distinguish between the cortical gray matter layer (with $\theta = 1$) from the deeper white matter (with $\theta = 0$).

Using MRIs of ferret brains, we created a computational model of the initial brain size and shape, which was discretized using tetrahedral meshes with over 1 million tetrahedral elements

**Table 1.** Human genetic variants modeled in ferret developmental brain phenotypes.

| Gene | Change in geometry | Relevant brain developmental disorder |
|---|---|---|
| *SCN3A* (*Smith et al., 2018*) | Excessive number of tightly packed folds | Polymicrogyria |
| *ASPM* (*Johnson et al., 2018*) | Reduced brain size and cortical surface area | Microcephaly |
| Cdk5 (*Shinmyo et al., 2017*) | Reduced depth of the sulcus | Lissencephaly |
| *ARHGAP11B* (*Kalebic et al., 2018*) | Expansion in both the radial and tangential dimensions | Megalencephaly |
| *TMEM161B* (*Akula et al., 2023b*) | Reduced gyrus size and sulcal depth | Lissencephaly |

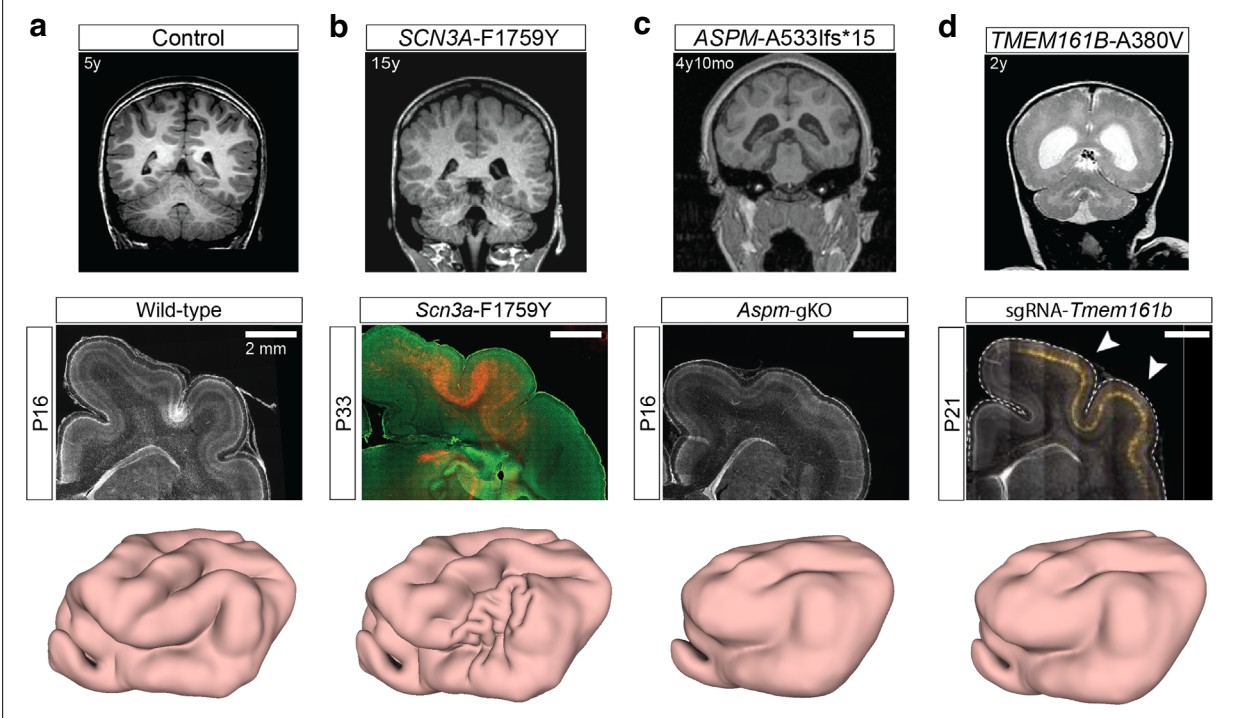

**Figure 6.** Modeling malformations of cortical development (MCD) using our model. (**a**) Control, (**b**) *SCN3A*, (**c**) *ASPM*, (**d**) *TMEM161B*. For each example, we show human (top) and ferret (middle) brain MRIs (images adapted from *Smith et al., 2018*; *Johnson et al., 2018*; *Akula et al., 2023b*). We then perform a modified numerical brain simulation on the P8 model with different tangential growth rate and cortical thickness parameters, including (**a**) the original growth rate and cortical thickness, (**b**) a reduction of the cortical thickness at a localized zone, (**c**) a reduction of the growth rate globally, and (**d**) a reduction of the growth rate and an increase of the cortical thickness globally. All numerical simulation results (bottom) qualitatively capture the cortical malformations.

using Netgen (*NGSolve Team, 2019*). Using a finite element method implemented using a discretized version of the energy of the system (1), we minimized the energy by quasistatic equilibration using an explicit solver (*Tallinen et al., 2016*), while growth was applied incrementally using the form described earlier by expanding the tetrahedral elements with inversion handling (*Stomakhin et al., 2012*) and a nodal pressure formulation (*Bonet and Burton, 1998*). Self-avoidance of the surface was handled using the penalty-based vertex-triangle contact processing (*Ericson, 2004*). We also enforce the condition that there is no growth in the central part as well as in the bottom part of the brain to better simulate the development of ferret brains. See Appendix 1 for more details.

## Malformations of cortical development

In *Figure 6*, we showed various human and ferret brain MRIs with MCD. The *SCN3A* MRI images were adapted from *Smith et al., 2018* (human: 15 years; ferret: P33). The *ASPM* MRI images were adapted from *Johnson et al., 2018* (human: 4 years 10 months; ferret: P16). The *TMEM161B* MRI images were adapted from *Akula et al., 2023b* (human: 2 years; ferret: P21).

The numerical simulation results for cortical malformations were obtained using the computational model described above, with the cortical thickness and growth rate parameters modified either locally or globally. See Appendix 1 for more details.

## Code availability statement

Computer codes for numerical simulations and morphometric analyses are available on GitHub at https://github.com/garyptchoi/ferret-brain-morphogenesis (copy archived at *Choi, 2025*).

## Acknowledgements

We thank Roberto Toro and Katja Heuer for the ferret brain data, and Jun Young Chung and James Weaver for their help with preliminary experiments. GPTC and LM are supported in part by the Harvard Quantitative Biology Initiative and the NSF-Simons Center for Mathematical and Statistical Analysis of Biology at Harvard, award no. 1764269. GPTC is also supported by the CUHK Faculty of Science Direct Grant for Research (Project Code 4053650). RSS is supported by R00NS112604 and R01NS140046. CAW is supported by the NINDS through R01NS032457 and R37NS035129, grant 62587 from the John Templeton Foundation (the opinions expressed in this publication are those of the authors and do not necessarily reflect the views of the John Templeton Foundation), and by the Allen Discovery Center for Human Brain Evolution, a Paul G Allen Frontiers Group advised program of the Paul G Allen Family Foundation. CAW is an Investigator of the Howard Hughes Medical Institute. LM is also supported by the Simons Foundation and the Henri Seydoux Fund.

## Additional information

### Funding

| Funder | Grant reference number | Author |
|---|---|---|
| Harvard Quantitative Biology Initiative and the NSF-Simons Center for Mathematical and Statistical Analysis of Biology at Harvard | 1764269 | Gary PT Choi L Mahadevan |
| Simons Foundation | | L Mahadevan |
| Henri Seydoux Fund | | L Mahadevan |
| CUHK Faculty of Science Direct Grant for Research | 4053650 | Gary PT Choi |
| National Institute of Neurological Disorders and Stroke | R01NS032457 | Christopher A Walsh |
| National Institute of Neurological Disorders and Stroke | R37NS035129 | Christopher A Walsh |
| John Templeton Foundation | 62587 | Christopher A Walsh |
| Allen Discovery Center | | Christopher A Walsh |
| National Institute of Neurological Disorders and Stroke | R00NS112604 | Richard S Smith |
| National Institute of Neurological Disorders and Stroke | R01NS140046 | Richard S Smith |

The funders had no role in study design, data collection and interpretation, or the decision to submit the work for publication.

### Author contributions

Gary PT Choi, Conceptualization, Software, Formal analysis, Validation, Investigation, Visualization, Methodology, Writing – original draft, Writing – review and editing; Chunzi Liu, Formal analysis, Validation, Investigation, Visualization, Methodology, Writing – original draft, Writing – review and editing;

Sifan Yin, Software, Formal analysis, Validation, Investigation, Visualization, Methodology, Writing – original draft, Writing – review and editing; Gabrielle Séjourné, Data curation, Formal analysis, Investigation, Methodology, Writing – original draft, Writing – review and editing; Richard S Smith, Data curation, Formal analysis, Investigation, Visualization, Methodology, Writing – review and editing; Christopher A Walsh, Conceptualization, Supervision, Funding acquisition, Investigation, Methodology, Writing – review and editing; L Mahadevan, Conceptualization, Formal analysis, Supervision, Funding acquisition, Investigation, Methodology, Project administration, Writing – review and editing

#### Author ORCIDs
Gary PT Choi http://orcid.org/0000-0001-5407-9111
Sifan Yin http://orcid.org/0000-0002-0296-3981
L Mahadevan https://orcid.org/0000-0002-5114-0519

Reviewer #1 (Public review): https://doi.org/10.7554/eLife.107141.3.sa1
Reviewer #2 (Public review): https://doi.org/10.7554/eLife.107141.3.sa2
Author response https://doi.org/10.7554/eLife.107141.3.sa3

## Additional files

#### Supplementary files
MDAR checklist

#### Data availability
Requests for the ferret brain data should be made to *Toro et al., 2018b*. All other data are included in the article and/or supplementary material.

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

## Appendix 1

### Physical gel model for ferret brain folding

#### Surface reconstruction

Surfaces of initial (pre-swelling) brain states were reconstructed from T2-weighted motion-corrected anatomical MR images of ferret brains at various ages (*Toro et al., 2018a*; *Toro et al., 2018b*) by R. Toro and K. Heuer. The reconstruction can be achieved by first computing masks of the pial and inner cortical surfaces from MRI data using the Nilearn Python module to generate and threshold image histograms (*Abraham et al., 2014*), and then utilizing the Brainbox web application (*Toro, 2017*) to manually correct region of interest (ROI) selections and convert masks from Nifti (.nii.gz) files to three-dimensional triangular meshes, with conservative corrections to the output meshes including smoothing and inversion of inward-pointing normal vectors applied, with further mesh corrections performed using Meshlab.

#### Swelling experiment

In *Figure 2*, we presented the swelling experimental results for an input P8 gel model and an input P16 gel model. In *Appendix 1—figure 1*, we present the swelling experiment for the P4 ferret brain with the same experimental setup, where the two-layer PDMS model was immersed in n-hexane for 1.5 hours. It can be observed that while the sulcal pattern is not particularly prominent in the input P4 gel model, complex folding patterns can be observed in the resulting swollen gel.

We remark that comparing the P4 swelling result with the P8 and P16 gel brain experiments in *Figure 2*, one can see that the folding patterns are notably different. A possible reason is that all gel models were immersed in n-hexane for swelling with the same duration (1.5 hours) regardless of their size and shape, which might result in the variation in the final gyrification patterns.

**Before swelling (P4)** 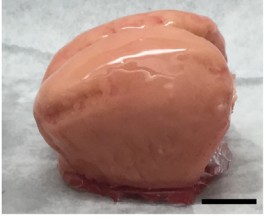 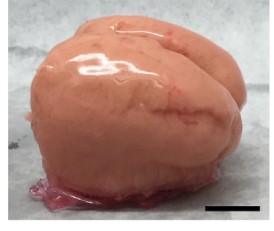 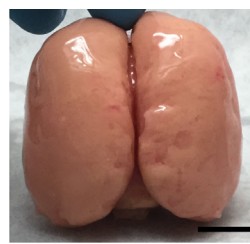

**After swelling** 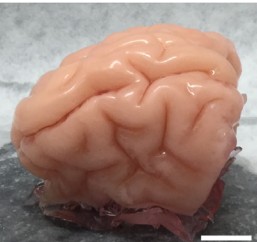 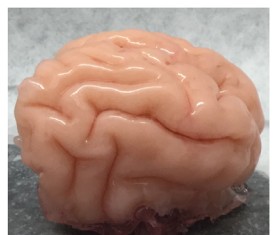 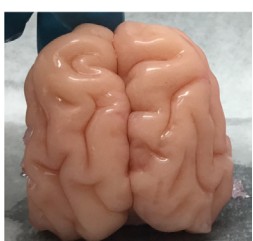

**Appendix 1—figure 1.** Swelling experiment for an input P4 gel brain. Here, the two-layer PDMS model of a P4 ferret brain was immersed in n-hexane for 1.5 hours. Analogous to the gel brain experiments presented in the main text, gyrification patterns can be observed in the swollen gel brain. Scale bar = 1 cm.

### Computational model for ferret brain folding

#### Formulation

For the numerical simulation of ferret brain development, we follow the approach in *Tallinen et al., 2014*; *Tallinen et al., 2016* and consider a material consisting of a layer of gray matter on top of a deep layer of white matter. The material is assumed to be neo-Hookean with volumetric strain energy density

$$W = \frac{\mu}{2}\left[Tr(\mathbf{F}\mathbf{F}^T)J^{-2/3} - 3\right] + \frac{K}{2}(J-1)^2, \tag{2}$$

where $\mathbf{F}$ is the deformation gradient, $J = \det(\mathbf{F})$, μ is the shear modulus, and $K$ is the bulk modulus. We assume that $K = 5\mu$ for a modestly compressible material.

Three geometrical parameters of the 3D brain models are the brain size $R$, the cortical thickness $T$, and the tangential expansion $g^2$. For ferret brain development, we follow the empirical scaling law for gray matter volume to thickness and set $R/T \approx 10$ with the tangential expansion ratio $g \approx 1.9$. An indicator function

$$\theta(y) = \frac{1}{1 + e^{10(y/T-1)}} \tag{3}$$

is applied for distinguishing between the cortical layer (with $\theta = 1$) and white matter zone (with $\theta = 0$). Here, $y$ is the distance from surface in material coordinates.

## Numerical experiments

Tetrahedral meshes were generated from the MRI-based reconstructed surfaces using the Netgen software (*NGSolve Team, 2019*), with each of them consisting of over 1 million tetrahedral elements. The numerical simulation procedure was implemented in C++. The energy of the system was minimized by quasistatic equilibration using an explicit scheme. Growth was applied by expanding the tetrahedral elements with inversion handling (*Stomakhin et al., 2012*) and nodal pressure formulation (*Bonet and Burton, 1998*). We further set the innermost part and the bottom part of the brain tetrahedral meshes to be non-growing regions to better simulate the development of ferret brains. See *Tallinen et al., 2014*; *Tallinen et al., 2016* for more details of the computation setup.

We simulated the stepwise growth of the ferret brain from P0 to P4, from P4 to P8, from P8 to P16, and from P16 to P32. *Appendix 1—figure 2* shows different views of the simulation results.

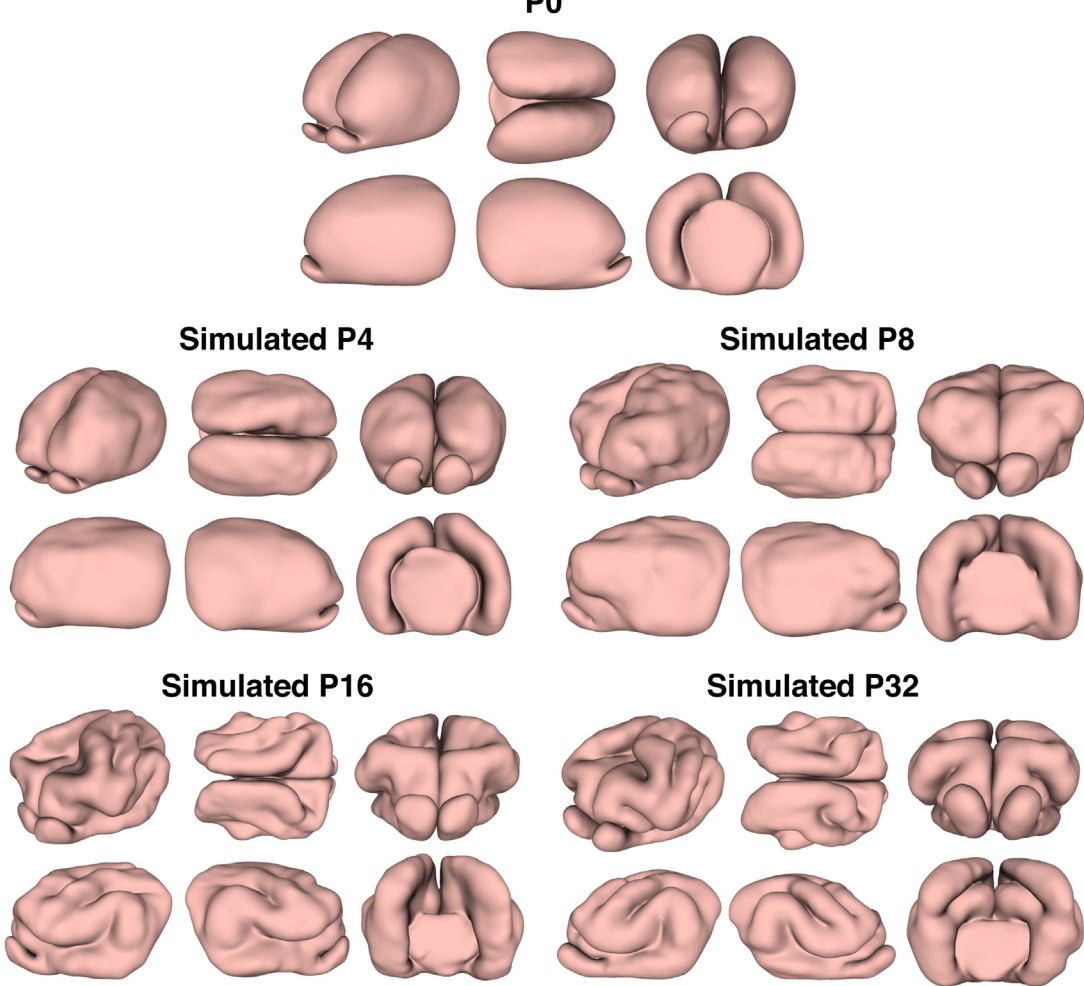

**Appendix 1—figure 2.** Stepwise growth simulation of ferret brain. The P0 brain and the simulated brains from P0 to P4, from P4 to P8, from P8 to P16, and from P16 to P32 are shown. For each brain, six different views are provided (not to scale).

## Mesh independence

One may ask about the effect of mesh size on the numerical simulation. *Appendix 1—figure 3* shows the continuous simulation results starting from P0 with different mesh resolutions. It can be observed that the results are similar, which shows the mesh independence of the numerical simulation.

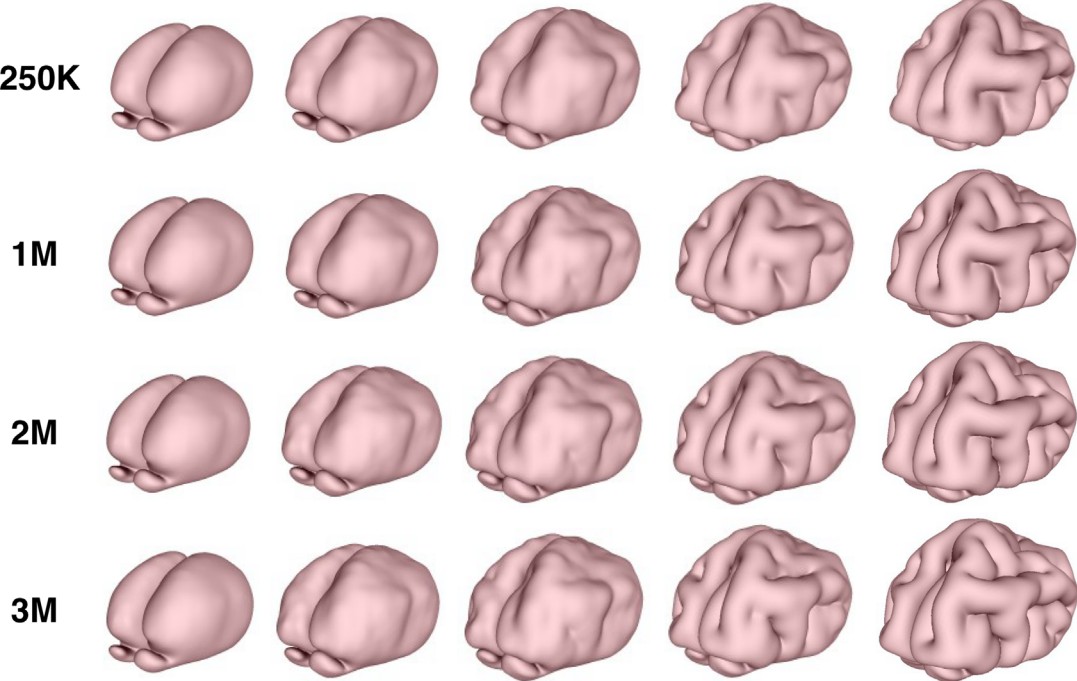

**Appendix 1—figure 3.** Mesh independence for the numerical simulation. The continuous growth simulations from P0 to adolescence using P0 brain tetrahedral meshes with 250,000 tets (first row), 1 million tets (second row), 2 million tets (third row), and 3 million tets (fourth row). It can be observed that the resulting folding patterns are highly similar.

## Morphometric analysis

### Analyzing the brain surfaces using surface parameterization

The FLASH method (*Choi et al., 2015*) was used for quantifying the similarity of the numerically simulated brain and the MRI-based brain. Denote the two brain surfaces to be compared by $\mathcal{M}$ and $\mathcal{N}$. The FLASH method produces two bijective spherical mappings $f : \mathcal{M} \to \mathbb{S}^2$ and $g : \mathcal{N} \to \mathbb{S}^2$, so that both surfaces are mapped to the unit sphere with prescribed landmark pairs on them optimally aligned. Here, $g$ is a conformal map from $\mathcal{N}$ to the unit sphere $\mathbb{S}^2$, and $f$ is a landmark-constrained optimized conformal map from $\mathcal{M}$ to $\mathbb{S}^2$. More specifically, $f$ achieves a balance between the landmark mismatch error and the conformality distortion, with the following combined energy minimized:

$$E(f) = \int_{\mathcal{M}} \|\nabla f\|^2 + \eta \sum_{i=1}^{t} \|f(p_i) - g(q_i)\|^2, \tag{4}$$

where $\{p_i, q_i\}$ are corresponding landmark curves on $\mathcal{M}$ and $\mathcal{N}$, respectively, and $\eta$ is a balancing parameter. The coronal sulcus (cns), suprasylvian sulcus (sss), presylvian sulcus (prs), and pseudosylvian sulcus (pss) on both the left and right hemispheres of the ferret brain are used as the landmark curves to ensure that the two spherical parameterizations are consistent. We set $\eta = 10$ to achieve an accurate alignment of the two spherical parameterizations without inducing a large conformal distortion.

After obtaining the spherical parameterizations with the landmarks optimally aligned, we can compare the folding patterns of the two brain surfaces by evaluating the similarity of their shape indices on $\mathbb{S}^2$. The shape index is a dimensionless surface measure defined based on the surface mean curvature and the surface Gaussian curvature (*Koenderink and van Doorn, 1992*). Specifically, the mean curvature $H$ and Gaussian curvature $K$ of any surface $\mathcal{S}$ are given by

$$H(v) = \frac{1}{2}(\kappa_1(v) + \kappa_2(v)) \tag{5}$$

and

$$K(v) = \kappa_1(v)\kappa_2(v), \tag{6}$$

where $v$ is a point on $\mathcal{S}$, and $\kappa_1(v), \kappa_2(v)$ are the principal curvatures at $v$. The shape index $I(v)$ (**Koenderink and van Doorn, 1992**) at each point $v$ is then given by

$$I(v) = \frac{2}{\pi}\arctan\left(\frac{H(v)}{\sqrt{(H(v))^2 - K(v)}}\right). \tag{7}$$

It is easy to see that $I(v) \in [-1, 1]$ for any $v$. Intuitively, for brain cortical surfaces, the sulci are with a smaller value of $I$ and the gyri are with a larger value of $I$.

Now, with the aid of the spherical parameterizations, we can easily evaluate the similarity of the shape index distributions of two brain surfaces. Specifically, the similarity $s$ of the shape index distributions $I_\mathcal{M}, I_\mathcal{N}$ of the two surfaces $\mathcal{M}, \mathcal{N}$ is defined as

$$s(\mathcal{M}, \mathcal{N}) = 1 - \frac{1}{2m}\sum_{k=1}^{m}\left|I_\mathcal{M}(v_k) - I_\mathcal{N}((g^{-1} \circ f)(v_k))\right|, \tag{8}$$

where $\{v_k\}_{k=1}^{m}$ are the vertices of $\mathcal{M}$. In particular, since $I_\mathcal{M}, I_\mathcal{N} \in [-1, 1]$, we have

$$0 \le \left|I_\mathcal{M}(v_k) - I_\mathcal{N}((g^{-1} \circ f)(v_k))\right| \le 2 \tag{9}$$

for all $k$. Consequently, we have $s \in [0, 1]$, and $s = 1$ if the two brain surfaces are identical. As discussed in the main text, our result of $s \approx 0.83$ shows that the real and simulated brain surfaces are highly similar.

Besides analyzing the entire brain surfaces, one can also quantify the shape difference for each half-brain surface using the mapping and shape similarity quantification method and different norms as described in **Yin et al., 2025**:

$$s(\mathcal{M}, \mathcal{N}) = 1 - \frac{1}{2m^{1/p}}\left\|I_\mathcal{M}(v_k) - I_\mathcal{N}((g^{-1} \circ f)(v_k))\right\|_p, \tag{10}$$

where $\|\cdot\|_p$ is the $p$-norm. As shown by the quantification results in **Appendix 1—table 1**, the real and simulated half-brain surfaces are highly similar. These additional results again demonstrate the effectiveness of our model for simulating ferret brain folding.

**Appendix 1—table 1.** Similarity between the real and simulated ferret half-brain surfaces.

| Surface | Similarity (1-norm) | Similarity (2-norm) | Similarity ($\infty$-norm) |
|---|---|---|---|
| Real P8 and simulated P8 (left) | 0.8040 | 0.7432 | 1.0000 |
| Real P8 and simulated P8 (right) | 0.7893 | 0.7260 | 1.0000 |
| Real P16 and simulated P16 (left) | 0.8247 | 0.7632 | 1.0000 |
| Real P16 and simulated P16 (right) | 0.7980 | 0.7339 | 1.0000 |

## Spherical harmonics representations

With the spherical parameterizations, we further computed the spherical harmonics representation of the two surfaces with different maximum order $N$ used. Denote $(\theta, \phi)$ as the spherical coordinates of $\mathbb{S}^2$, where $\theta \in [0, \pi]$ is the elevation angle and $\phi \in [-\pi, \pi]$ is the azimuth angle. The spherical harmonics functions $Y_n^m : [0, \pi] \times [-\pi, \pi] \to \mathbb{R}$ of order $n$ and degree $m$ are given as follows **Müller, 2006**:

$$Y_n^m(\theta, \phi) = \begin{cases} (-1)^{|m|}\sqrt{2}K_n^m \sin(|m|\phi)P_n^{|m|}(\cos\theta) & : m < 0, \\ (-1)^{|m|}\sqrt{2}K_n^m \cos(m\phi)P_n^m(\cos\theta) & : m > 0, \\ K_n^0 P_n^0(\cos\theta) & : m = 0, \end{cases} \tag{11}$$

where

$$K_n^m = \sqrt{\frac{(2n+1)(n-|m|)!}{4\pi(n+|m|)!}},$$

(12)

and

$$P_n^m(x) = \frac{(-1)^m}{2^n n!}(1-x^2)^{m/2}\frac{d^{n+m}}{dx^{n+m}}(x^2-1)^n.$$

(13)

With the maximum order $N$ prescribed, we can consider all $Y_n^m$ with $n = 0, 1, \ldots, N$ and $m = -n, -n+1, \ldots, 0, \ldots, n-1, n$. We can then approximate any given surface as a linear combination of this set of SH functions:

$$\begin{cases} x(\theta, \phi) \approx \sum_{n=0}^{N}\sum_{m=-n}^{n} a_n^m Y_n^m(\theta, \phi), \\ y(\theta, \phi) \approx \sum_{n=0}^{N}\sum_{m=-n}^{n} b_n^m Y_n^m(\theta, \phi), \\ z(\theta, \phi) \approx \sum_{n=0}^{N}\sum_{m=-n}^{n} c_n^m Y_n^m(\theta, \phi), \end{cases}$$

(14)

where the coefficients $a_n^m, b_n^m, c_n^m$ can be computed from the spherical parameterization (see *Brechbühler et al., 1995* for details). As the maximum order $N$ increases, more geometric details of the surface can be captured in the spherical harmonics representation.

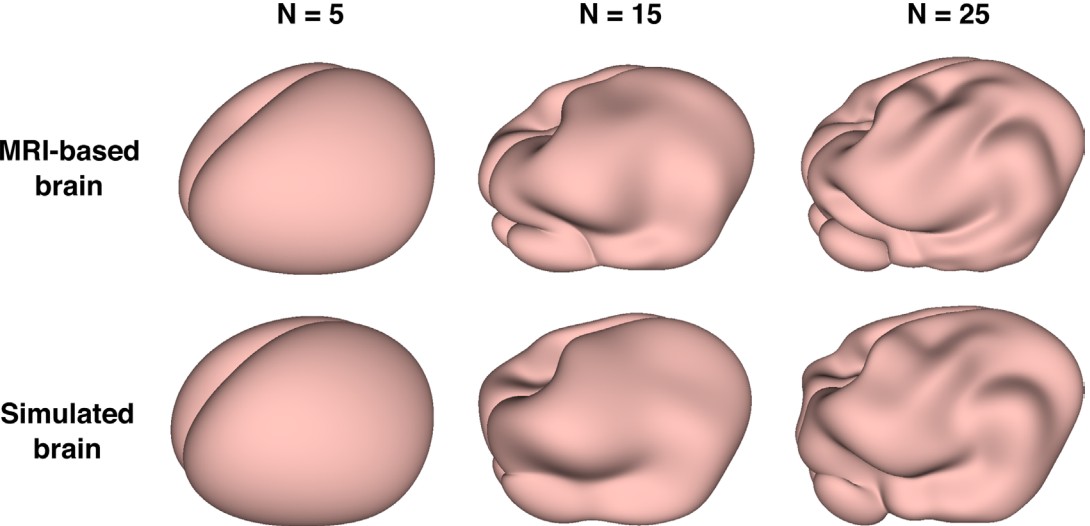

**Appendix 1—figure 4.** The spherical harmonics representations of the MRI-based brain and the numerically simulated brain with different maximum order $N$ used.

From *Appendix 1—figure 4*, it can be observed that for different maximum order $N$, the spherical harmonics representations of MRI-based brain and the numerically simulated brain match very well. More specifically, when $N = 5$ is used, both representations give an overall smooth brain shape without folding. When $N = 15$ is used, both of them start to exhibit some folding at consistent locations. When $N = 25$ is used, both of them show highly similar sulci and gyri patterns. This again indicates that the two brain surfaces are geometrically similar.

## Cortical malformations

In the main text, we considered various types of cortical malformations and discussed how our proposed model can be modified to capture their features qualitatively. In this section, additional analyses are provided.

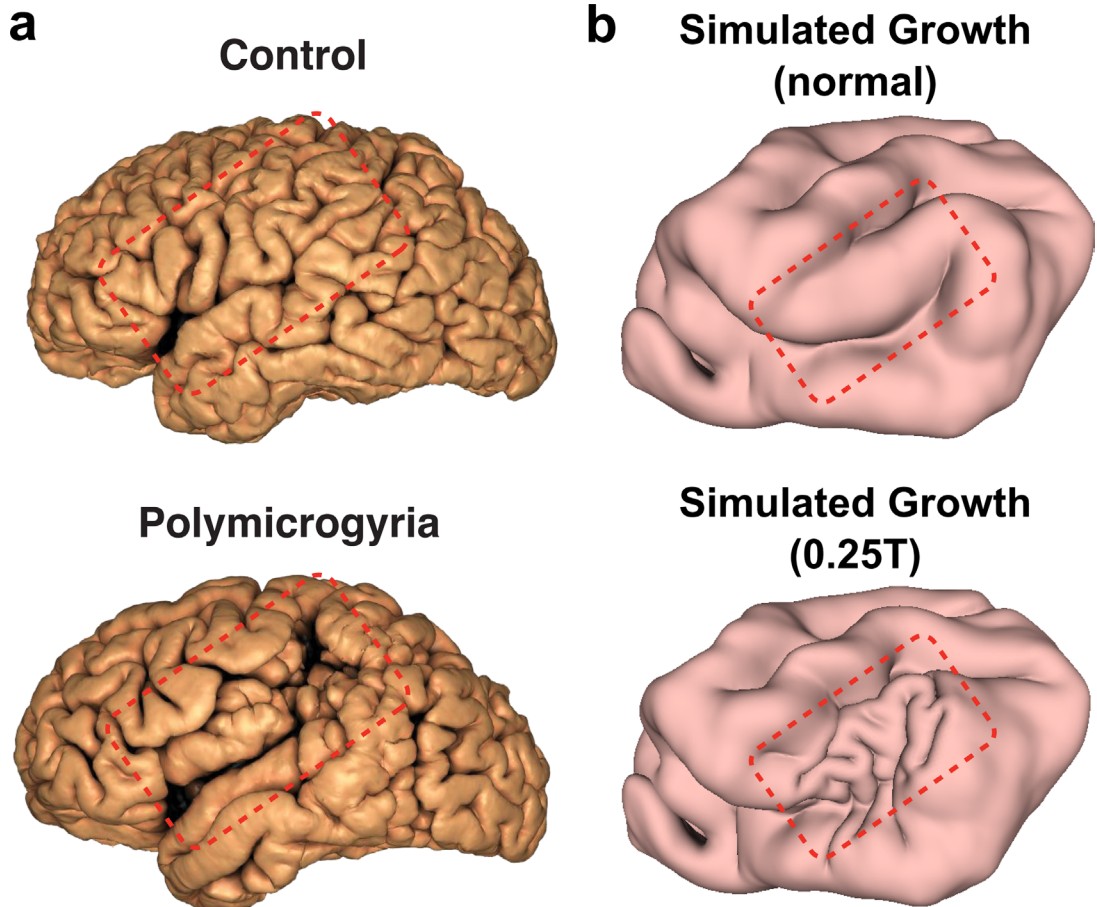

**Appendix 1—figure 5.** Comparison between human brain surface MRI reconstructions and our ferret brain simulations for the malformations of cortical development (MCD) polymicrogyria (PMG). (**a**) Surface MRI reconstructions of human MRIs performed using FreeSurfer software of control (top) and age-matched affected individual with a gain-of-function *SCN3A* variant resulting in PMG. Red box outlines the PMG of perisylvian and surrounding areas, where the normal gyri and sulci form microgyri/sulci, pushing together to make an appearance of "Moroccan leather".) The numerical growth simulations on the P8 ferret brain without modification in the cortical thickness (top) and with a reduced cortical thickness to 1/4 of the original thickness, i.e., $0.25T$ (bottom). The differences are highlighted by the red boxes.

© 2018 Elsevier Inc. Images in panel a are reprinted from *Smith et al., 2018* with permission. They are not covered by the CC-BY 4.0 licence and further reproduction of this panel would need permission from the copyright holder.

In *Appendix 1—figure 5*, we provide a comparison between human brain surface MRI reconstructions and our ferret brain simulations for the MCD PMG. Here, the human brain surface reconstruction images (*SCN3A* case and control age-matched subject) are adapted from *Smith et al., 2018*, where MRI stacks were processed to extract the surfaces of the gray/white matter and gray matter/cerebrospinal fluid boundaries, and to define the cortical surface using FreeSurfer software. It can be observed that using our computational model with a modification in the cortical thickness, we obtain ferret brain simulation results with a good qualitative match.

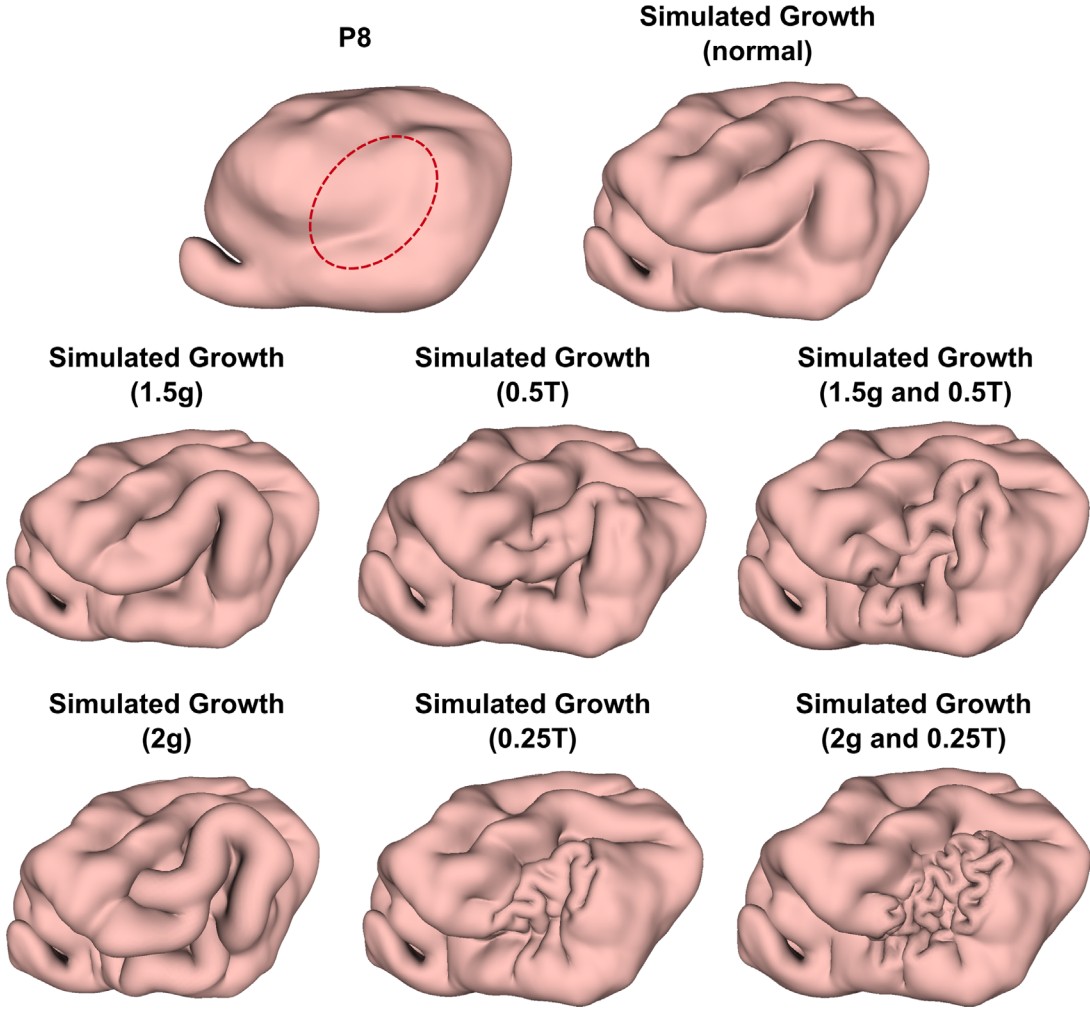

**Appendix 1—figure 6.** Numerical simulations with locally modified brain growth. We performed numerical simulations on the P8 brain with different modifications in the tangential growth rate $g$ and the cortical layer thickness $T$ at a localized region (dotted ellipse). The results without modification (normal), with an increased growth rate to 1.5 times the original growth rate ($1.5g$) or 2 times the original growth rate ($2g$), with a reduced cortical thickness to $1/2$ of the original thickness ($0.5T$) or $1/4$ of the original thickness ($0.25T$), and with a modification in both the growth rate and the cortical thickness ($1.5g$ and $0.5T$ $2g$ and $0.25T$) are presented.

In *Appendix 1—figure 6*, we further performed numerical simulations on the P8 ferret brain with different modifications in the tangential growth rate and cortical thickness at a localized region (dotted ellipse). Specifically, we first considered increasing the tangential growth rate at the localized region to 1.5 times or 2 times the original rate. In the numerical simulation results, more prominent folds can be observed at the corresponding region. Also, we considered reducing the cortical thickness to 1/2 or 1/4 of the original thickness at the localized region. In the simulation results, we can see that new folds are created at the localized region. Finally, we considered modifying both the tangential growth rate and the cortical thickness. In this case, a higher level of abnormal gyrification can be observed locally in the simulation results. Altogether, the modified brain experiments show that different cortical malformations can be effectively simulated by modifying the tangential growth rate and cortical thickness.

