## [Editor Report · eLife Assessment]

This **important** study characterizes the morphogenesis of cortical folding in the ferret and human cerebral cortex using complementary physical and computational modeling. Notably, these approaches are applied to charting, in the ferret model, known abnormalities of cortical folding in humans. The study finds **convincing** evidence that variation in cortical thickness and expansion accounts for deviations in morphology and supports these findings using cutting-edge approaches from both physical gel models and numerical simulations. The study will be of broad interest to the field of developmental neuroscience.

---

## [Referee Report · Reviewer #1 (Public review)]

The manuscript by Choi and colleagues investigates the impact of variation in cortical geometry and growth on cortical surface morphology. Specifically, the study uses physical gel models and computational models to evaluate the impact of varying specific features/parameters of the cortical surface. The study makes use of this approach to address the topic of malformations of cortical development and finds that cortical thickness and cortical expansion rate are the drivers of differences in morphogenesis.

The study is composed of two main sections. First, the authors validate numerical simulation and gel model approaches against real cortical postnatal development in the ferret. Next, the study turns to modelling malformations in cortical development using modified tangential growth rate and cortical thickness parameters in numerical simulations. The findings investigate three genetically linked cortical malformations observed in the human brain to demonstrate the impact of the two physical parameters on folding in the ferret brain.

This is a tightly presented study that demonstrates a key insight into cortical morphogenesis and the impact of deviations from normal development. The dual physical and computational modeling approach offers the potential for unique insights into mechanisms driving malformations. This study establishes a strong foundation for further work directly probing the development of cortical folding in the ferret brain.

---

## [Referee Report · Reviewer #2 (Public review)]

Summary:

Based on MRI data of the ferret (a gyrencephalic non-primate animal, in whom folding happens postnatally), the authors create in vitro physical gel models and in silico numerical simulations of typical cortical gyrification. They then use genetic manipulations of animal models to demonstrate that cortical thickness and expansion rate are primary drivers of atypical morphogenesis. These observations are then used to explain cortical malformations in humans.

Strengths:

The paper is very interesting and original, and combines physical gel experiments, numerical simulations, as well as observations in MCD. The figures are informative, and the results appear to have good overall face validity.

Comment on the revised version from the Reviewing Editor:

The reviewers are happy with the authors replies and the eLife Assessment has been amended accordingly.

---

## [Author Response]

The following is the authors’ response to the original reviews.

**Reviewer #1 (Public review):**
The manuscript by Choi and colleagues investigates the impact of variation in cortical geometry and growth on cortical surface morphology. Specifically, the study uses physical gel models and computational models to evaluate the impact of varying specific features/parameters of the cortical surface. The study makes use of this approach to address the topic of malformations of cortical development and finds that cortical thickness and cortical expansion rate are the drivers of differences in morphogenesis.The study is composed of two main sections. First, the authors validate numerical simulation and gel model approaches against real cortical postnatal development in the ferret. Next, the study turns to modelling malformations in cortical development using modified tangential growth rate and cortical thickness parameters in numerical simulations. The findings investigate three genetically linked cortical malformations observed in the human brain to demonstrate the impact of the two physical parameters on folding in the ferret brain.This is a tightly presented study that demonstrates a key insight into cortical morphogenesis and the impact of deviations from normal development. The dual physical and computational modeling approach offers the potential for unique insights into mechanisms driving malformations. This study establishes a strong foundation for further work directly probing the development of cortical folding in the ferret brain. One weakness of the current study is that the interpretation of the results in the context of human cortical development is at present indirect, as the modelling results are solely derived from the ferret. However, these modelling approaches demonstrate proof of concept for investigating related alterations more directly in future work through similar approaches to models of the human cerebral cortex.

We thank the reviewer for the very positive comments. While the current gel and organismal experiments focus on the ferret only, we want to emphasize that our analysis does consider previous observations of human brains and morphologies therein (Tallinen et al., Proc. Natl. Acad. Sci. 2014; Tallinen et al., Nat. Phys. 2016), which we compare and explain. This allows us to analyze the implications of our study broadly to understand the explanations of cortical malformations in humans using the ferret to motivate our study. Further analysis of normal human brain growth using computational and physical gel models can be found in our companion paper (Yin et al., 2025), now also published to eLife: S. Yin, C. Liu, G. P. T. Choi, Y. Jung, K. Heuer, R. Toro, L. Mahadevan, Morphogenesis and morphometry of brain folding patterns across species. eLife, 14, RP107138, 2025. doi:10.7554/eLife.107138

In future work, we plan to obtain malformed human cortical surface data, which would allow us to further investigate related alterations more directly. We have added a remark on this in the revised manuscript (please see page 8–9).

**Reviewer 2 (Public review):**
Summary:Based on MRI data of the ferret (a gyrencephalic non-primate animal, in whom folding happens postnatally), the authors create in vitro physical gel models and in silico numerical simulations of typical cortical gyrification. They then use genetic manipulations of animal models to demonstrate that cortical thickness and expansion rate are primary drivers of atypical morphogenesis. These observations are then used to explain cortical malformations in humans.Strengths:The paper is very interesting and original, and combines physical gel experiments, numerical simulations, as well as observations in MCD. The figures are informative, and the results appear to have good overall face validity.

We thank the reviewer for the very positive comments.

Weaknesses:On the other hand, I perceived some lack of quantitative analyses in the different experiments, and currently, there seems to be rather a visual/qualitative interpretation of the different processes and their similarities/differences. Ideally, the authors also quantify local/pointwise surface expansion in the physical and simulation experiments, to more directly compare these processes. Time courses of eg, cortical curvature changes, could also be plotted and compared for those experiments. I had a similar impression about the comparisons between simulation results and human MRI data. Again, face validity appears high, but the comparison appeared mainly qualitative.

We thank the reviewer for the comments. Besides the visual and qualitative comparisons between the models, we would like to point out that we have included the quantification of the shape difference between the real and simulated ferret brain models via spherical parameterization and the curvature-based shape index as detailed in main text Fig. 4 and SI Section 3. We have also utilized spherical harmonics representations for the comparison between the real and simulated ferret brains at different maximum order N. In our revision, we have included more calculations for the comparison between the real and simulated ferret brains at more time points in the SI (please see SI page 6). As for the comparison between the malformation simulation results and human MRI data in the current work, since the human MRI data are two-dimensional while our computational models are threedimensional, we focus on the qualitative comparison between them. In future work, we plan to obtain malformed human cortical surface data, from which we can then perform the parameterization-based and curvature-based shape analysis for a more quantitative assessment.

I felt that MCDs could have been better contextualized in the introduction.

We thank the reviewer for the comment. In our revision, we have revised the description of MCDs in the introduction (please see page 2).

**Reviewer #1 (Recommendations for the authors):**
The study is beautifully presented and offers an excellent complement to the work presented by Yin et al. In its current form, the malformation portion of the study appears predominantly reliant on the numerical simulations rather than the gel model. It might be helpful, therefore, to further incorporate the results presented in Figure S5 into the main text, as this seems to be a clear application of the physical gel model to modelling malformations. Any additional use of the gel models in the malformation portion of the study would help to further justify the necessity and complementarity of the dual methodological approaches.

We thank the reviewer for the suggestion. We have moved Fig. S5 and the associated description to the main text in the revised manuscript (please see the newly added Figure 5 on page 6 and the description on page 5–7). In particular, we have included a new section on the physical gel and computational models for ferret cortical malformations right before the section on the neurology of ferret and human cortical malformations.

One additional consideration is that the analyses in the current study focus entirely on the ferret cortex. Given the emphasis in the title on the human brain, it may be worthwhile to either consider adding additional modelling of the human cortex or to consider modifying the title to more accurately align with the focus of the methods/results.

We thank the reviewer for the suggestion. While the current gel and organismal experiments focus on the ferret only, we want to emphasize that our analysis does consider previous observations of human brains and morphologies therein (Tallinen et al., Proc. Natl. Acad. Sci. 2014; Tallinen et al., Nat. Phys. 2016), which we compare and explain. This allows us to analyze the implications of our study broadly to understand the explanations of cortical malformations in humans using the ferret to motivate our study. Therefore, we think that the title of the paper seems reasonable. To further highlight the connection between the ferret brain simulations and human brain growth, we have included an additional comparison between human brain surface reconstructions adapted from a prior study and the ferret simulation results in the SI (please see SI Section S4 and SI Fig. S5 on page 9–10).

Two additional minor points:Table S1 seems sufficiently critical to the motivation for the study and organization of the results section to justify inclusion in the main text. Of course, I would leave any such minor changes to the discretion of the authors.

We thank the reviewer for the suggestion. We have moved Table S1 and the associated description to the main text in the revised manuscript (please see Table 1 on page 7).

Page 7, Column 1: “macacques” → “macaques”.

We thank the reviewer for pointing out the typo. We have fixed it in the revised manuscript (please see page 8).

**Reviewer #2 (Recommendations for the authors):**
The methods lack details on the human MRI data and patients.

We thank the reviewer for the comment. Note that the human MRI data and patients were from prior works (Smith et al., Neuron 2018; Johnson et al., Nature 2018; Akula et al., Proc. Natl. Acad. Sci. 2023) and were used for the discussion on cortical malformations in Fig. 6. In the revision, we have included a new subsection in the Methods section and provided more details and references of the MRI data and patients (please see page 9–10).